# FLAT-LoRA: LOW-RANK ADAPTION OVER A FLAT LOSS LANDSCAPE

## ABSTRACT

Fine-tuning large-scale pre-trained models is prohibitively expensive in terms of computational and memory costs. Low-Rank Adaptation (LoRA), a popular Parameter-Efficient Fine-Tuning (PEFT) method, provides an efficient way to fine-tune models by optimizing only a low-rank matrix. Despite recent progress made in improving LoRA's performance, the connection between the LoRA optimization space and the original full parameter space is often overlooked. A solution that appears flat in the LoRA space may exist sharp directions in the full parameter space, potentially harming generalization performance. In this paper, we propose Flat-LoRA, an efficient approach that seeks a low-rank adaptation located in a flat region of the full parameter space. Instead of relying on the well-established sharpness-aware minimization approach, which can incur significant computational and memory burdens, we utilize random weight perturbation with a Bayesian expectation loss objective to maintain training efficiency and design a refined perturbation generation strategy for improved performance. Experiments on natural language processing and image classification tasks with various architectures demonstrate the effectiveness of our approach.

## 1 INTRODUCTION

Pre-training followed by fine-tuning is a widely adopted training pipeline among modern machine learning practitioners for achieving state-of-the-art (SOTA) performance (Girshick et al., 2014; Kolesnikov et al., 2020; Wortsman et al., 2022; Yu et al., 2024b), leveraging the versatile knowledge within the pre-trained models. However, the enormous size of these pre-trained models makes fine-tuning all parameters for downstream tasks resource-intensive, making it impractical to store optimizer states or multiple model weights when dealing with multiple tasks. Recently, Low-Rank Adaptation (LoRA) (Hu et al., 2022) has been proposed to address this resource challenge. In LoRA fine-tuning, only a low-rank matrix is optimized and then added to the pre-trained weights after training, incurring no additional computational or memory costs during inference. This approach significantly reduces the number of trainable parameters, thereby lowering the training cost as well as storage cost when dealing with different tasks.

Many works have been proposed to enhance the performance of LoRA by introducing more dedicated budgets for rank allocation (Zhang et al., 2023a), decomposing optimization for direction and magnitude updates (Liu et al., 2024b), or designing better initialization strategy for LoRA parameters (Meng et al., 2024; Wang et al., 2024), etc. These studies demonstrate the significant potential for improving LoRA performance. However, the connection between the LoRA optimization space and the original full parameter space is often overlooked. Essentially, LoRA restricts training to a much lower-dimensional subspace, and its performance depends on the properties of the solutions within this subspace in relation to the full parameter space, as the merged weights are ul-

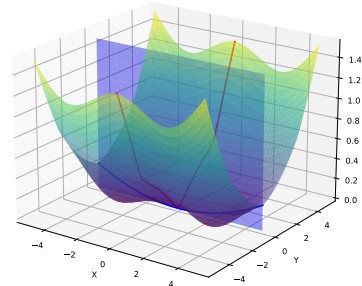

Figure 1: Illustration of LoRA optimization. LoRA constrains training in a lower-dimensional subspace (blue). A flat minima in LoRA subspace (blue curve) may exhibit sharp direction in full parameter space (red curve).

timately used during inference. As illustrated in Figure 1, a flat minima in the LoRA space (blue) may exhibit sharp direction (red) in the view of the full parameter space, which potentially degenerates the generation performance.

It is widely believed that minima with a flatter loss landscape can better adapt to distribution shifts between training and test datasets and lead to improved generalization performance (Hochreiter & Schmidhuber, 1994; 1997). This idea has given rise to a well-established training strategy called Sharpness-Aware Minimization (SAM), which has shown great generalization improvement in training neural networks. Applying SAM to large language models (LLMs)' training together with LoRA is certainly promising, but there are several issues should be discussed. First, unlike the existing attempts that flatten the landscape in a LoRA subspace (Li et al., 2024a), which is not aware of the sharpness outside the LoRA space, we pursue a solution that aligns with a flatter loss landscape in the full weight space. Second, the original SAM doubles the training time cost, which is impractical for fine-tuning large models. Additionally, to capture the sharpness of the full parameter space, we need to calculate the gradients and store the perturbations of the full weights, which contradicts the principles of parameter-efficient fine-tuning (PEFT). To cope with these challenges, we propose using random weight perturbations that do not require additional gradient steps and can be efficiently stored with random seeds to maintain time efficiency and memory, and design effective generation strategies to improve generalization performance.

Our main contribution can be summarized as follows:

- We propose Flat-LoRA that firstly aims to optimize the sharpness of the loss landscape within the full parameter space where the low-rank adaptation resides. It incurs minimal additional computational and memory costs and can be easily integrated with existing techniques to enhance LoRA performance, delivering consistent improvements.

- We propose to use expected Bayesian loss to optimize the sharpness for keeping the training efficiency and design effective generation strategy to generate random weight perturbation to enhance the generalization performance, making it easy for practical usage.

- Experiments on natural language processing and computer vision tasks with various scales of models to demonstrate that our approach can achieve state-of-the-art performance.

## 2 RELATED WORK

### 2.1 FLAT MINIMA AND GENERALIZATION

The connection between the flatness of local minima and generalization has received much attention (Hochreiter & Schmidhuber, 1997; Chaudhari et al., 2017; Keskar et al., 2017; Dinh et al., 2017; Izmailov et al., 2018; Li et al., 2018b; Wu et al., 2020). Recently, many works have tried to improve generalization by seeking flat minima (Tsuzuku et al., 2020; Zheng et al., 2021; Bisla et al., 2022). For example, Chaudhari et al. (2017) propose Entropy-SGD to search for flat regions by minimizing local entropy. Wen et al. (2018) design SmoothOut framework to smooth out the sharp minima. Notably, Sharpness-Aware Minimization (SAM) (Foret et al., 2020) establishes a generic training scheme for seeking flat minima by formulating a min-max problem and encourage parameters sitting in neighborhoods with uniformly low loss, achieving state-of-the-art generalization improvements across various tasks. However, SAM requires twice the training time as regular training, limiting its applications to large scale training.

Another branch of methods for recovering flat minima involves minimizing the expected Bayesian training loss under random weight perturbation (RWP), which is efficient and doesn't require additional gradient step (Bisla et al., 2022). Wang & Mao (2021) propose Gaussian model perturbation (GMP) as a regularization scheme for improving SGD training, but it remains inefficient for multiple for noise sampling. Bisla et al. (2022) connect the smoothness of loss objective to generalization and adopted filter-wise random Gaussian perturbation generation to recover flat minima and improve generalization. Li et al. (2024c) further enhance the generalization performance of RWP by introducing an adaptive perturbation generation strategy and a mixed loss objective. (Wu et al., 2022; Li et al., 2024b) demonstrate that injecting small random noise into LLMs before or during fine-tuning can improve generalization. However, when applying these approaches to PEFT training, we must be mindful of the additional memory and time costs they may introduce.

## 2.2 Low-rank Adaption and Variants

Recent works have indicated that the intrinsic dimension for optimizing deep neural networks (DNNs) may be significantly lower than the number of parameters (Li et al., 2018a; Gur-Ari et al., 2018; Wu et al., 2024). Notably, Li et al. (2022a) demonstrate that the training trajectory of DNNs can be low-dimensional and proposed subspace optimization to enhance training efficiency and robustness (Li et al., 2022b). Low-Rank Adaptation (LoRA) (Hu et al., 2022) is proposed to model the weight changes for each layer during fine-tuning. It effectively decreases the number of trainable parameters, thereby lowering the memory burden for training and storage. This approach is currently the mainstream because it avoids adding any overhead during inference while often demonstrating strong performance (Wang et al., 2023; Liu et al., 2024a).

Many works have been proposed to enhance the performance of LoRA. AdaLoRA (Zhang et al., 2023a) dynamically prunes insignificant weights during fine-tuning through singular value decomposition (SVD), enabling allocating more rank to important areas under a fixed parameter budget. DoRA (Liu et al., 2024b) enhances the model's expressiveness by introducing learnable magnitudes that decomposes optimization for direction and magnitude updates. LoRA+ (Hayou et al., 2024) proposes to use different learning rates for the two matrices in LoRA to improve convergence. PiSSA (Meng et al., 2024) proposes to use to SVD decomposition of the original matrix $\mathbf{W}$ to initialize the LoRA matrices, which provides a better initialization for LoRA parameters. LoRA-GA (Wang et al., 2024) proposes to approximate the gradient of the original matrix by performing SVD on sampled gradient and properly scaling the initialized matrices. LoRA-Pro (Wang & Liang, 2024) further proposes to align each gradient step to the full fine-tuning. Li et al. (2024a) consider applying SAM to LoRA parameters and develop a resource-efficient SAM, balancedness-aware regularization (BAR), tailored for scale-invariant problems such as fine-tuning language models with LoRA. In this paper, we improve LoRA by optimizing the sharpness of the full parameter space.

## 3 Method

In this section, we first give a brief review on the low-rank adaption (LoRA). We then introduce our LoRA optimization objective considering the flatness of the landscape. We finally describe our random perturbation generation strategy for effectively improving the generalization performance.

### 3.1 LoRA: Low-Rank Adaption

Based on the finding that DNNs' optimization happens in a subspace with much smaller dimensions than the number of parameters (Li et al., 2018a; 2022a), LoRA utilizes low-rank matrices to model the weight change for each layers' weights $\mathbf{W} \in \mathbb{R}^{m \times n}$ during the fine-tuning as $\Delta \mathbf{W} = \mathbf{BA}$, where $\mathbf{B} \in \mathbb{R}^{m \times r}$ and $\mathbf{A} \in \mathbb{R}^{r \times n}$ with the rank $r \ll \{m, n\}$ to achieve parameter efficiency. We omit the scaling factor $s = \alpha/r$ here for simplicity as it can be merged into $\mathbf{A}$ and $\mathbf{B}$. For the original output $\mathbf{h} = \mathbf{Wx}$, the modified forward pass is

$$\mathbf{h} = \mathbf{Wx} + \Delta \mathbf{Wx} = (\mathbf{W} + \mathbf{BA})\mathbf{x}. \tag{1}$$

At initialization, matrix $\mathbf{A}$ is commonly initialized with Kaiming distribution (He et al., 2015) and matrix $\mathbf{B}$ is set to zeros. During the training, only the low-rank matrices $\mathbf{A}$ and $\mathbf{B}$ are optimized with the pre-trained weight $\mathbf{W}$ being frozen. During the inference, the low-rank matrices $\Delta \mathbf{W}$ are merged to the pre-trained weight $\mathbf{W}$, and there is no additional computational or memory costs.

### 3.2 LoRA with a Flat Landscape

Despite recent efforts to improve LoRA performance, most studies focus solely on finding solutions performing well in the LoRA space, specifically the rank $r$ matrix space $\mathcal{M}_r = \{\Delta \mathbf{W} \in \mathbf{R}^{m \times n} \mid \text{rank}(\Delta \mathbf{W}) = r\}$. Let $f(\boldsymbol{x}; \mathbf{W})$ be a transformer, and $L(f(\boldsymbol{x}_i; \mathbf{W}), \boldsymbol{y}_i)$ denote the loss function ($L_i(\mathbf{W})$ for short; we focus on a single LoRA module). Given a dataset $\mathcal{S} = \{(\boldsymbol{x}_i, \boldsymbol{y}_i)\}$, the empirical training loss is defined as $L(\mathbf{W}) = \frac{1}{|\mathcal{S}|} \sum_{i=1}^{|\mathcal{S}|} L_i(\mathbf{W})$. Following the well-established sharpness-aware minimization (SAM) objective (Foret et al., 2020), Li et al. (2024a) apply SAM to LoRA parameters and study the scale-invariant properties of these parameters with SAM:

$$\min_{\mathbf{A}, \mathbf{B}} \max_{\|(\epsilon_\mathbf{A}, \epsilon_\mathbf{B})\| \le \rho} L\left(\mathbf{W} + (\mathbf{B} + \epsilon_\mathbf{B})(\mathbf{A} + \epsilon_\mathbf{A})\right), \tag{2}$$

where $L(\cdot)$ denotes the loss objective. However, focusing solely on the properties of the optimization space defined by LoRA parameters may have limitations. During inference, the low-rank adaption $\Delta\mathbf{W}$ is merged into the pre-trained weights $\mathbf{W}$. A solution that performs well within the LoRA space may be situated in a sharp region of the full parameter space, as illustrated in Figure 1, which could potentially harm overall generalization. To be more clear, the equivalent weight perturbation applied to $\mathbf{W}$ by Equ (2) is

$$\mathbf{B}\epsilon_{\mathbf{A}} + \epsilon_{\mathbf{B}}\mathbf{A} + \epsilon_{\mathbf{B}}\epsilon_{\mathbf{A}} = c\mathbf{B}\mathbf{B}^\top\nabla L(\mathbf{W}) + c\nabla L(\mathbf{W})\mathbf{A}^\top\mathbf{A} + c^2\nabla L(\mathbf{W})\mathbf{A}^\top\mathbf{B}^\top\nabla L(\mathbf{W}), \quad (3)$$

where $c = \rho/\sqrt{\|\mathbf{B}^\top\nabla L(\mathbf{W})\|^2 + \|\nabla L(\mathbf{W})\mathbf{A}^\top\|^2}$ is a scaling factor. One can see that the perturbation direction is not aligned with the direction $\nabla L(\mathbf{W})$, which maximizes the loss of the merged weights as in SAM. Notably, when $\mathbf{B}$ is initialized as zero as defaulted in Hu et al. (2022), $\mathbf{B}$ will remain small during the training (Hao et al., 2024) and Equ. (3) becomes:

$$\mathbf{B}\epsilon_{\mathbf{A}} + \epsilon_{\mathbf{B}}\mathbf{A} + \epsilon_{\mathbf{B}}\epsilon_{\mathbf{A}} \approx c\nabla L(\mathbf{W})\mathbf{A}^\top\mathbf{A}. \quad (4)$$

This means Equ (2) only optimizes the sharpness along the column space spanned by $\mathbf{A}$, which constitutes a small subspace of the full parameter space. As demonstrated in Table 5, solely applying SAM constraints on the LoRA parameters does not effectively improve the generalization.

Therefore, it is crucial to consider the loss landscape of $L(\mathbf{W} + \Delta\mathbf{W})$, and we need to find a low rank adaption $\Delta\mathbf{W}$ that positions the merged weights in a flat region of the full parameter space. Our flat loss objective can be formulated as follows:

$$\min_{\mathbf{A},\mathbf{B}} \max_{\|\epsilon\|\leq\rho} L(\mathbf{W} + \mathbf{B}\mathbf{A} + \epsilon). \quad (5)$$

However, directly applying SAM to optimize the sharpness of the merged weight space has several disadvantages: 1) it doubles the training cost, which is less desirable with large models, and 2) it requires storing an additional copy of weights for perturbation, which contradicts the principle of parameter-efficient fine-tuning. To achieve a flatter loss landscape while maintaining time and memory efficiency, we propose relaxing the maximization problem in Eq. (5) to an expectation, resulting in the following Bayesian expected loss objective:

$$\min_{\mathbf{A},\mathbf{B}} \mathbb{E}_{\epsilon\sim\mathcal{N}(0,\sigma^2\mathbf{I})} L(\mathbf{W} + \mathbf{B}\mathbf{A} + \epsilon), \quad (6)$$

where $\sigma$ controls the variance magnitude of the noise, which we will describe in the next section. This expected loss can be seen as applying a smoothing filter over the loss landscape within the full parameter space, and optimizing it can help recover flatter minima (Bisla et al., 2022). For each optimization step, we would sample a noise $\epsilon$ and calculate the perturbed gradient to optimize the low-rank matrices $\mathbf{A}$ and $\mathbf{B}$. Note that the noise is generated based on the model weights, thus incurring no additional gradient steps as SAM does.

### 3.3 EFFECTIVE RANDOM PERTURBATION GENERATION

We then describe how to effectively generate random weight perturbation, which are essential for optimizing sharpness and enhancing generalization performance. Let $\mathbf{W}' = \mathbf{W} + \mathbf{B}\mathbf{A}$. For the merged weight $\mathbf{W}' \in \mathbb{R}^{m\times n}$ that represents a linear layer with input dimension $n$ and output dimension $m$, our design considers the following two perspectives:

- Filter structure: we aim to generate the weight noise by filter (Bisla et al., 2022). There contains $m$ filters $\mathbf{W}' = (\mathbf{W}'_{1,:}, \mathbf{W}'_{2,:}, \cdots, \mathbf{W}'_{m,:})$ that process the input $\mathbf{x} \in \mathbb{R}^n$. Elements within a filter of larger norm should receive a larger strength of perturbation.
- Input dimension: we hope that the variance introduced to the forward pass by the added random weight perturbation is independent of the input dimension. Given an input dimension $n$, the magnitude of noise added to each element should be scaled by a factor of $1/\sqrt{n}$.

Finally, our random weight generation scheme is formulated as follows:

$$\epsilon \sim \mathcal{N}\left(0, \frac{\sigma^2}{n}\text{diag}(\|\mathbf{W}'_{1,:}\|_2^2, \|\mathbf{W}'_{2,:}\|_2^2, \cdots, \|\mathbf{W}'_{m,:}\|_2^2)\mathbf{I}_{m\times n}\right), \quad (7)$$

where $\mathbf{I}_{m\times n}$ denotes a matrix of size $m \times n$ with all ones. Here $\sigma$ is the hyper-parameter that needs to be selected for controlling the perturbation strength. An overview of LoRA and our Flat-LoRA is illustrated in Figure 2.

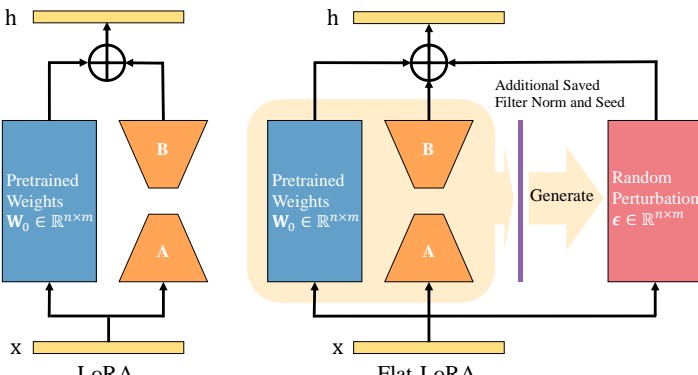

Figure 2: **Illustration of LoRA (Left) and Flat-LoRA (Right).** Flat-LoRA, building upon LoRA, optimizes the sharpness of the merged weights in the full parameter space by adding designed random weight perturbations. It does not require extra gradient steps as SAM and remains memory efficient by only storing the random seed and few filter norms, which takes less than 2‰ of the trainable parameters used by LoRA.

**Analysis on the variance of the activation.** We then analyze the effects of introducing random weight perturbation on the activation. Given an input $\mathbf{x} \in \mathbb{R}^n$, and under the hypothesis that $\mathbf{x}$ is a random vector where each element has the same variance $\text{var}[\mathbf{x}_i]$ and expectation $\mathbb{E}[\mathbf{x}_i]$, we have:

$$\text{var}[\mathbf{W}'_{j,:}\mathbf{x}] = \|\mathbf{W}'_{j,:}\|_2^2 \cdot \text{var}[\mathbf{x}_i]. \tag{8}$$

After injecting random weight perturbation $\epsilon$, we have:

$$\text{var}[(\mathbf{W}' + \epsilon)_{j,:}\mathbf{x}] = \|\mathbf{W}'_{j,:}\|_2^2 \cdot \text{var}[\mathbf{x}_i] + \text{var}[\epsilon_{j,:}\mathbf{x}] \tag{9}$$

$$= \|\mathbf{W}'_{j,:}\|_2^2 \cdot \text{var}[\mathbf{x}_i] + \sum_{i=1}^{n} \text{var}\left[\epsilon_{j,i}\mathbf{x}_i\right] \tag{10}$$

$$= \|\mathbf{W}'_{j,:}\|_2^2 \cdot \text{var}[\mathbf{x}_i] + n \cdot \frac{\sigma^2}{n} \|\mathbf{W}'_{j,:}\|_2 \cdot \left(\text{var}[\mathbf{x}_i] + \mathbb{E}^2[\mathbf{x}_i]\right) \tag{11}$$

$$= (1 + \sigma^2)\|\mathbf{W}'_{j,:}\|_2^2 \cdot \text{var}[\mathbf{x}_i] + \sigma^2 \|\mathbf{W}'_{j,:}\|_2 \cdot \mathbb{E}^2[\mathbf{x}_i]. \tag{12}$$

Thus, by injecting random weight perturbations $\epsilon$, we introduce variance into the forward activation with a rate of $\sigma^2$ along with a bias term determined by the expectation of $\mathbf{x}_i$. Note that, since we introduce a scaling factor of $1/n$ for the variance in noise generation (i.e., Equ. (7)), the resulting increased variance is independent of the input dimension $n$. This increased variance helps escape from sharp local minima. Additionally, we note that this variance would not increase exponentially during the forward propagation of the network due to the existence of layer normalization.

**Storing random seed for memory efficiency.** Memory cost is an important factor to consider for PEFT training. To optimize Eqn. (6), we first generate random perturbation $\epsilon$ and then perform gradient descent with $\nabla L(\mathbf{W} + \mathbf{BA} + \epsilon)$. Thus, we need to store the weight perturbation for recovering the weight after obtaining the perturbed gradient. When model is large, storing a copy of weight perturbation is prohibitive. Luckily, for random weight perturbation, we only need to store the seed for random generator and corresponding norms for each filter $\|\mathbf{W}'_{1,:}\|_2^2, \|\mathbf{W}'_{2,:}\|_2^2, \cdots, \|\mathbf{W}'_{m,:}\|_2^2$, allowing us to recover the random perturbation $\epsilon$ when necessary. This approach incurs minimal additional memory and offers significant advantages over SAM, which requires calculating the full gradient, thereby necessitating a hard copy of the perturbation that cannot be reduced.

**An easier approach for mixed precision training.** When mixed precision training is used, which is commonly adopted for large-scale training, we have an easier approach to seamlessly integrate the perturbation injection process into the precision casting, introducing no additional memory cost. Specifically, in mixed-precision training, two copies of model weights are maintained in memory: the full-precision FP32 weights and the half-precision FP/BF16 weights. We can inject random weight perturbation during the half-precision auto-cast step before the forward pass, thus eliminating the need to store a copy of the weight perturbation or the filter norms. However, our main approach

is to efficiently store the perturbation based on filter norms and random seed, which is more general and does not require mixed-precision training.

# 4 EXPERIMENTS

In this section, we evaluate the performance of Flat-LoRA on various benchmark tasks. We first conduct experiments on natural language understanding tasks using a subset of GLUE datasets (Wang et al., 2019b) with T5-base model (Raffel et al., 2020). We then experiment over image classification tasks with CLIP ViT-B/32 model (Radford et al., 2021). Subsequently, we evaluate mathematical reasoning and coding abilities using the Llama 2-7B model (Touvron et al., 2023). We finally give ablation studies and discussions on our method. The code is attached in the supplement materials.

## 4.1 EXPERIMENTS ON NATURAL LANGUAGE UNDERSTANDING

**Setting.** We finetune T5-Base model on several datasets from GLUE benchmark, including MNLI, SST, CoLA, QNLI, and MRPC, following (Wang et al., 2024). Performance is evaluated on the development set using accuracy as the primary metric. We use LoRA with rank 8 and 16 with LoRA alpha 16. We finetune the models with 10 epochs with a cosine learning rate schedule, except for MNLI and QNLI we use 1 epochs. We use learning rate of 0.0005 for LoRA fine-tuning and 0.0001 for full fine-tuning with weight decay 0.1. The random perturbation strength $\sigma$ is set to 0.05 with an cosine increasing strategy. Mean and standard deviations are calculated over 3 independent trials.

**Results.** As shown in Table 1, Flat-LoRA consistently outperforms LoRA for ranks 8 and 16, achieving average performance gains of 0.34% and 0.56%, respectively. In some cases, the performance of LoRA does not improve or even deteriorate when increasing the rank from 8 to 16, as seen with the CoLA and MRPC datasets, which are relatively small and susceptible to overfitting. Flat-LoRA effectively addresses the overfitting issue and achieves greater improvements with increasing LoRA rank, demonstrating the advantages of our flat loss objective.

Table 1: Results (%) on fine-tuning T5-base with a subset of GLUE datasets.

| Method | MNLI | SST2 | CoLA | QNLI | MRPC | Avg. |
|---|---|---|---|---|---|---|
| Full FT | $86.19_{\pm0.04}$ | $94.15_{\pm0.09}$ | $82.84_{\pm0.12}$ | $93.10_{\pm0.04}$ | $89.22_{\pm0.23}$ | 89.10 |
| LoRA ($r=8$) | $\mathbf{86.24}_{\pm0.02}$ | $94.55_{\pm0.07}$ | $82.87_{\pm0.22}$ | $93.06_{\pm0.03}$ | $88.97_{\pm0.42}$ | 89.13 |
| Flat-LoRA ($r=8$) | $86.20_{\pm0.04}$ | $\mathbf{94.75}_{\pm0.20}$ | $\mathbf{83.61}_{\pm0.38}$ | $\mathbf{93.16}_{\pm0.09}$ | $\mathbf{89.59}_{\pm0.37}$ | $\mathbf{89.47}$ |
| LoRA ($r=16$) | $86.49_{\pm0.06}$ | $94.52_{\pm0.21}$ | $82.89_{\pm0.44}$ | $92.97_{\pm0.05}$ | $88.89_{\pm0.44}$ | 89.15 |
| Flat-LoRA ($r=16$) | $\mathbf{86.51}_{\pm0.01}$ | $\mathbf{94.84}_{\pm0.02}$ | $\mathbf{84.08}_{\pm0.31}$ | $\mathbf{93.28}_{\pm0.03}$ | $\mathbf{89.83}_{\pm0.34}$ | $\mathbf{89.71}$ |

## 4.2 EXPERIMENTS ON IMAGE CLASSIFICATION

**Setting.** We finetune the CLIP-ViT-B/32 model on five image classification tasks, including CIFAR-10/100 (Krizhevsky & Hinton, 2009), Cars (Krause et al., 2013), SVHN (Netzer et al., 2011), and DTD (Cimpoi et al., 2014). We resize all input image to a size of $224 \times 224$ and freeze the classification head. We try LoRA with rank 8 and 16 and finetune the models with 10 epochs with a cosine annealing schedule. The learning rate is set to 0.0005 for LoRA and $1 \times 10^{-5}$ for full fine-tuning with weight decay 0.1. The random perturbation strength $\sigma$ is set to 0.15 with an cosine increasing strategy. Mean and standard deviations are calculated over 3 independent trials.

**Results.** We measure the performance with classification accuracy and report the results in Table 2. We observe that Flat-LoRA consistently outperforms LoRA with ranks 8 and 16, showing average improvements of 0.56% and 0.74%, respectively. Notably, Flat-LoRA with rank 8 surpasses both LoRA with rank 16 and full fine-tuning by 0.28%. These results confirm the effectiveness of our flat loss objective on improving LoRA performance.

## 4.3 RESULTS ON LLAMA-2

**Setting.** To evaluate the scalability of Flat-LoRA, we fine-tune Llama-2-7B (Touvron et al., 2023) on two tasks: *math* and *code*. We use a learning rate of $5e-4$ and cosine learning rate scheduler

Table 2: Results (%) on fine-tuning CLIP ViT-B/32 with image classification datasets.

| Method | CIFAR-10 | CIFAR-100 | Cars | SVHN | DTD | Avg. |
|---|---|---|---|---|---|---|
| Full FT | $97.99_{\pm0.01}$ | $89.06_{\pm0.11}$ | $73.30_{\pm0.43}$ | $97.44_{\pm0.03}$ | $76.80_{\pm0.25}$ | 86.92 |
| LoRA ($r = 8$) | $97.90_{\pm0.02}$ | $87.74_{\pm0.13}$ | $73.22_{\pm0.53}$ | $97.49_{\pm0.08}$ | $76.86_{\pm0.34}$ | 86.64 |
| Flat-LoRA ($r = 8$) | $\mathbf{98.09}_{\pm0.04}$ | $\mathbf{88.64}_{\pm0.23}$ | $\mathbf{74.17}_{\pm0.71}$ | $\mathbf{97.59}_{\pm0.04}$ | $\mathbf{77.51}_{\pm0.28}$ | **87.20** |
| LoRA ($r = 16$) | $97.99_{\pm0.03}$ | $88.12_{\pm0.23}$ | $73.80_{\pm0.42}$ | $97.56_{\pm0.08}$ | $77.34_{\pm0.32}$ | 86.92 |
| Flat-LoRA ($r = 16$) | $\mathbf{98.21}_{\pm0.04}$ | $\mathbf{89.27}_{\pm0.07}$ | $\mathbf{74.89}_{\pm0.52}$ | $\mathbf{97.71}_{\pm0.10}$ | $\mathbf{78.24}_{\pm0.44}$ | **87.66** |

with a warmup ratio of 0.03. We use LoRA with rank 8 and alpha 16 and the training epoch is set to 2. Following Wang et al. (2024), the backbone of Lllma 2-7B uses BF16 precision and the parameters of LoRA modules use FP32 precision for better performance. For *math* task, we finetune the model on MetaMathQA (Yu et al., 2024a) and evaluate it on GSM8K evaluation set (Cobbe et al., 2021). For *code* task, we finetune the model on Code-Feedback (Zheng et al., 2024) and evaluate it on HumanEval (Chen et al., 2021). We only use 100k training subsets for both tasks. The random perturbation strength $\sigma$ is set to 0.10. We also fine-tune a Llama 2-13B on the Alpaca dataset[1](Taori et al., 2023) and evaluate it on InstructEval(Chia et al., 2023), an instruction following benchmark. The experimental setting is set to the same as that in Ren et al. (2024), and the model is evaluated with the official code[2] provided by Chia et al. (2023).

**Results.** We measure the performance of the *math* task by accuracy and the *code* task by PASS@1 metric. From the results in Table 3, we observe that Flat-LoRA significantly enhances LoRA's performance under large-scale fine-tuning scenarios, achieving an improvement of +3.18% on the GSM8K dataset and 1.37% on the Human-Eval dataset. It is important to note that here our LoRA performance is much stronger than the results reported in previous works, e.g., 57.47% (ours) v.s. 42.08% (Wang et al., 2024) on GSM8K. Still, Flat-LoRA continues to demonstrate significant accuracy improvements over the baseline approach, highlighting the effectiveness of pursuing the flatness of the full parameter space when fine-tuning large LLM models.

Table 3: Results (%) on fine-tuning Llama-2-7B with GSM8K and Human-Eval datasets.

| Method | GSM8K | Human-Eval |
|---|---|---|
| Full FT | $59.36_{\pm0.85}$ | $35.31_{\pm2.13}$ |
| LoRA ($r = 8$) | $57.47_{\pm0.35}$ | $24.85_{\pm0.52}$ |
| Flat-LoRA ($r = 8$) | $\mathbf{60.65}_{\pm0.23}$ | $\mathbf{26.22}_{\pm0.79}$ |

We then focus on instruct-following tasks. From the results in Table 4, we observe that Flat-LoRA also consistently outperforms LoRA. We find that the improvements on DROP and Human-Eval are more pronounced (+0.71% and +1.83%, respectively), suggesting that flatter minima may better support math-related and coding-related tasks. This observation aligns with the findings in Table 3.

Table 4: Results on instruct-following tasks. We fine-tune Llama-2-13B model on Alpaca and evaluate InstructEval metrics.

| Method | MMLU | DROP | BBH | Human-Eval |
|---|---|---|---|---|
| LoRA ($r = 8$) | 51.42 | 37.57 | 34.72 | 13.41 |
| Flat-LoRA ($r = 8$) | **51.98** | **38.28** | **34.84** | **15.24** |

## 4.4 RESULTS ON STABLE DIFFUSION

**Setting.** Following the setting in DoRA (Liu et al., 2024b), we finetune SDXL (Podell et al., 2023) with the pipeline of Dreambooth (Ruiz et al., 2023) and the popular scripts implemented by HuggingFace [3]. The finetuning dataset, 3D Icons[4], contains 23 training images, all of which have a

---

[1] https://huggingface.co/datasets/yahma/alpaca-cleaned
[2] https://github.com/declare-lab/instruct-eval
[3] https://github.com/huggingface/diffusers/blob/main/examples/dreambooth/README_sdxl.md
[4] https://huggingface.co/datasets/linoyts/3d_icon

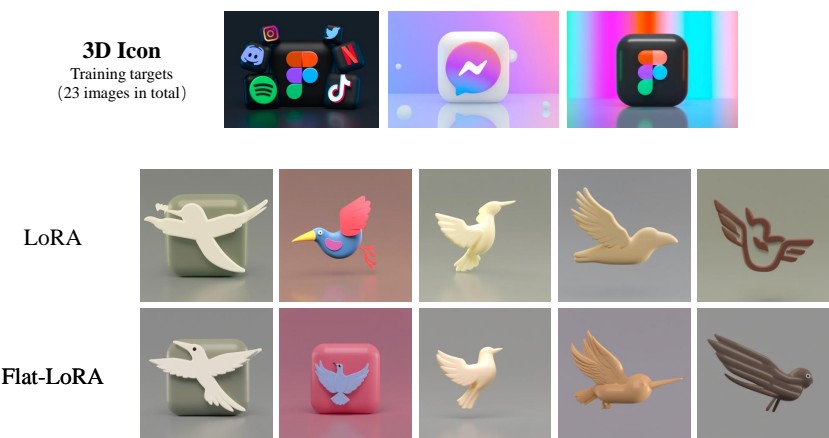

3D Icon
Training targets
(23 images in total)

LoRA

Flat-LoRA

Prompt: a TOK icon of a flying bird, in the style of TOK

Figure 3: Images generated with SDXL finetuning with LoRA and Flat-LoRA on the 3D icon datasets. The images of the same column are generated with the same seed for fair comparisons.

square. We finetune the model for 500 steps with a constant learning rate of 0.0001. The batch size is set to 1. The LoRA rank and alpha are set to 4. The $\sigma$ of Flat-LoRA is set to 0.1. Other hyperparameters are set to default values.

**Results.** As shown in Figure 3, Flat-LoRA exhibits better personalization than LoRA while maintaining better generation ability. For instance, in the second column, the image generated by Flat-LoRA includes a distinctive square behind the bird, aligning more closely with the "icon" feature present in the training images (top row). Furthermore, Flat-LoRA more effectively preserves the concept of eyes, whereas in columns 1, 3, and 5, the birds generated by LoRA are missing eyes.

## 4.5 COMPARISON WITH OTHER METHODS

We then compare our approach with other recently proposed methods for improving LoRA, including initialization-based methods such as PiSSA and LoRA-GA, as well as optimization-based methods like DoRA and LoRA+. Our experiments are conducted on the CoLA and MRPC datasets using the T5-base model with LoRA rank 8. The results are presented in Table 5. We observe that Flat-LoRA consistently outperforms previous methods by 0.53%. Furthermore, our flat loss objective can be easily integrated with earlier approaches to yield consistent improvements by 0.31% to 0.93%. This highlights the effectiveness of considering the sharpness of the full parameter space.

Table 5: Comparison with other methods on GLUE subsets using T5-Base.

| Method | CoLA | MRPC |
|---|---|---|
| LoRA (Hu et al., 2022) | $82.87_{\pm 0.22}$ | $88.03_{\pm 0.14}$ |
| PiSSA (Meng et al., 2024) | $83.18_{\pm 0.24}$ | $88.96_{\pm 0.44}$ |
| LoRA-GA (Wang et al., 2024) | $81.83_{\pm 0.21}$ | $87.58_{\pm 0.41}$ |
| DoRA (Liu et al., 2024b) | $83.16_{\pm 0.15}$ | $89.46_{\pm 0.37}$ |
| AdaLoRA (Zhang et al., 2023b) | $82.58_{\pm 0.56}$ | $88.29_{\pm 0.33}$ |
| DyLoRA (Valipour et al., 2023) | $82.98_{\pm 0.34}$ | $87.88_{\pm 0.42}$ |
| LoRA+ (Hayou et al., 2024) | $81.65_{\pm 0.34}$ | $89.30_{\pm 0.47}$ |
| Flat-LoRA (ours) | $\mathbf{83.61}_{\pm 0.38}$ | $89.59_{\pm 0.37}$ |
| Flat-PiSSA (ours) | $83.51_{\pm 0.48}$ | $89.89_{\pm 0.71}$ |
| Flat-LoRA-GA (ours) | $82.23_{\pm 0.34}$ | $88.15_{\pm 0.54}$ |
| Flat-DoRA (ours) | $83.56_{\pm 0.27}$ | $\mathbf{89.99}_{\pm 0.47}$ |
| Flat-LoRA+ (ours) | $82.56_{\pm 0.23}$ | $89.61_{\pm 0.44}$ |

In this paper, we adopt a stronger training baseline, including employing a larger learning rate and longer training epochs, which achieves significantly better performance than the results reported

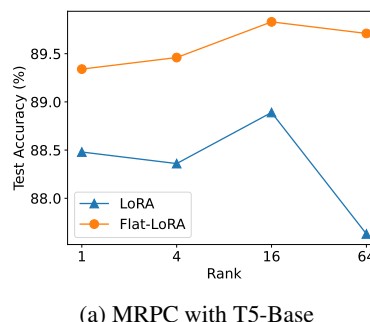 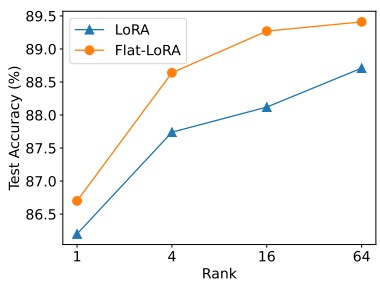

(a) MRPC with T5-Base  (b) CIFAR-100 with ViT-B/32

Figure 4: Performance comparison under different LoRA ranks. We keep LoRA alpha to 16 and vary the LoRA ranks among $\{1, 4, 16, 64\}$. Experiments are averaged with three independent trials.

in previous work (Wang et al., 2024). In fact, CoLA and MRPC are two datasets that achieve the most significant improvement by LoRA-GA as reported in the original paper (Wang et al., 2024). Under our experimental settings, LoRA-GA does not exhibit advantages over vanilla LoRA and can perform worse. This may be because LoRA-GA adopts a smart initialization strategy by maximizing gradient alignment with full parameter training, allowing for quicker convergence to a good local optimum (e.g., in just one epoch). However, such an initialization strategy may not be optimal for reaching a global optimum and exhibit unstable when the learning rate is large.

## 4.6 ABLATION STUDIES AND DISCUSSION

**Results under different LoRA ranks.** Following the settings in Section 4.1 and 4.2, we evaluate the performance of Flat-LoRA under different LoRA ranks. The results are shown in Figure 4. We observe that Flat-LoRA consistently outperforms LoRA across different LoRA ranks by +1.10% on MRPC and +1.15% on CIFAR-100. Even at LoRA rank 1, which is typically underfitting, Flat-LoRA still delivers a significant performance boost over LoRA. This highlights the importance of considering the sharpness of the full parameter space. Additionally, as the LoRA rank increases, we observe that LoRA's performance can degrade due to overfitting, particularly on MRPC, which is a small dataset with 3.7k data points. Flat-LoRA effectively mitigates this overfitting issue by identifying flatter minima that generalize better. Thus, we conclude that Flat-LoRA enhances LoRA fine-tuning performance not only in underfitting scenarios, where the rank is low and limited information from the full parameter space is explored, but also in high LoRA rank situations, where the risk of overfitting is more pronounced.

**Comparison with SAM.** We then compare Flat-LoRA with standard sharpness-aware minimization approach. Specifically, we consider applying SAM to the full parameter space, i.e., $\mathbf{W}$, and LoRA parameters $\mathbf{A}, \mathbf{B}$. We follow the settings in Section 4.1 and 4.2 and select the perturbation radius $\rho$ among $\{0.01, 0.05, 0.1, 0.2, 0.3, 0.5\}$, where $\rho = 0.05$ attains the best performance. From the results in Table 6, we observe that applying SAM on $\mathbf{W}$ achieves considerable better performance that on $\mathbf{A}, \mathbf{B}$, by +1.12% on CoLA and 1.31% on MRPC. However, applying SAM on $\mathbf{W}$ requires an additional memory of $\mathcal{O}(m \times n)$ for storing the adversarial weight perturbation, which can cause out-of-memory problem for fine-tuning large models. We also observe that directly applying SAM to $\mathbf{A}, \mathbf{B}$ does not bring performance improvement over vanilla LoRA, perhaps due to the maximum problem in SAM's optimization target is too strict for the LoRA subspace. Then for Flat-LoRA, we observe that it can achieve comparable or even better performance than LoRA with SAM applied on

Table 6: Comparison with SAM on GLUE subsets using T5-Base.

| Method | Flat Space | CoLA | MRPC | Additional Memory | Training time |
|---|---|---|---|---|---|
| LoRA | - | $82.87_{\pm 0.59}$ | $88.03_{\pm 0.14}$ | - | $1\times$ |
| LoRA+SAM | $\mathbf{A}, \mathbf{B}$ | $82.55_{\pm 0.49}$ | $87.65_{\pm 0.69}$ | $\mathcal{O}((m+n) \times r)$ | $2\times$ |
| LoRA+SAM | $\mathbf{W}$ | $\mathbf{83.67}_{\pm 0.39}$ | $88.56_{\pm 0.23}$ | $\mathcal{O}(m \times n)$ | $2\times$ |
| Flat-LoRA | $\mathbf{A}, \mathbf{B}$ | $83.19_{\pm 0.70}$ | $88.81_{\pm 0.51}$ | $\mathcal{O}(m+r)$ | $1\times$ |
| Flat-LoRA | $\mathbf{W}$ | $83.61_{\pm 0.38}$ | $\mathbf{89.59}_{\pm 0.37}$ | $\mathcal{O}(m)$ | $1\times$ |

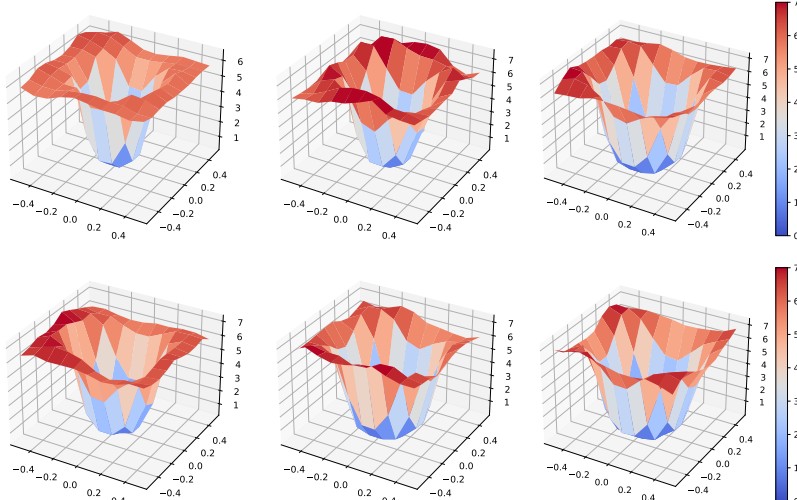

Figure 5: **Loss landscape visualization** with different LoRA ranks: 1 (**Left**) and 16 (**Middle**), and Full FT (Right), as well as different LoRA approaches: LoRA (**Up**) and Flat-LoRA (**Down**). Models are fine-tuned on CIFAR-100 with CLIP ViT-B/32.

**W**, but with minor additonal memory burden, e.g. $\mathcal{O}(m)$. Finally, it is worth note that Flat-LoRA maintains training efficiency as vanilla LoRA, where SAM approaches require doubled training time due to the extra gradient step involved.

**Landscape visualization.** In Figure 5, we plot the loss landscape of the merged weights of LoRA and Flat-LoRA with different loRA ranks. Following the plotting technique in (Li et al., 2018b), we uniformly sample $11 \times 11$ grid points in the range of $[-0.5, 0.5]$ from random "filter-normalized" direction. We observe that Flat-LoRA consistently achieves a flatter loss landscape than LoRA in both LoRA fine-tuning and full fine-tuning scenarios. An interesting observation is that when the LoRA rank is small, the loss landscape of the merged weight space tends to be sharper, highlighting the importance of considering the sharpness of the full parameter space when utilizing LoRA fine-tuning. Our Flat-LoRA enables a flat loss landscape comparable to full fine-tuning with a low LoRA rank. For instance, Flat-LoRA with a rank of 16 achieves a similarly flat landscape and obtains comparable performance to full fine-tuning.

## 5 CONCLUSION

In this paper, we introduce Flat-LoRA, an efficient low-rank adaptation approach that aims to optimize the sharpness of the loss landscape within the full parameter space that LoRA situates in. Deviating from standard sharpness-aware approach that incurs significant computation and memory burdens, we employ a Bayesian expectation loss objective minima and utilize designed random weight perturbations to pursuit flat minima, maintaining the training speed and memory efficiency characteristic of parameter-efficient fine-tuning. Flat-LoRA achieves state-of-the-art performance in LoRA fine-tuning and can be easily integrated with previous methods for consistent improvements. Extensive experiments on natural language processing and computer vision tasks with various scales of models demonstrate the effectiveness of our approach.

**Limitation and Future works.** One limitation of this paper is that we only consider fine-tuning and optimizing the sharpness of linear layers in transformer model. This approach is the common practice in fine-tuning LLMs for downstream tasks (Hu et al., 2022), and the linear layers account for the majority of model parameters (e.g. $> 99\%$). Future works could explore optimizing the sharpness of LayerNorm parameters, as our initial experiments in Appendix A have shown promising results. Additionally, since we can inject random weight perturbations during the `autocast` in mixed-precision training, our approach holds promise for enhancing low-bit training performance. Seeking flat minima during LoRA fine-tuning is also promising for reducing the forgetting of pre-trained knowledge. It is also promising to design more delicate noise generation strategy to enhance the generalization performance and improve the noise generation efficiency.

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

## A    EXTENDING PERTURBATION TO ALL LAYERS

We extend the injection of random weight perturbation to all layers, referred to as "Flat-LoRA (all)". Specifically, we additionally add perturbations to layernorm layers, biases, and class embeddings, etc. We generate noise based on the absolute weight $|\mathbf{W}|$. From the results in Table A1, we observe that Flat-LoRA (all) indeed improves performance, though the improvement is not as large as Flat-LoRA (Linear) over LoRA.

Table A1: Results on CIFAR-10/100 with CLIP ViT-B/32.

| Method | CIFAR-10 | CIFAR-100 |
|---|---|---|
| LoRA | $97.90_{\pm 0.02}$ | $87.74_{\pm 0.13}$ |
| Flat-LoRA (linear) | $98.09_{\pm 0.04}$ | $88.64_{\pm 0.23}$ |
| Flat-LoRA (all) | $\mathbf{98.13}_{\pm 0.03}$ | $\mathbf{88.76}_{\pm 0.19}$ |

## B    COMPARISON WITH OTHER SAM'S VARIANTS

We further compare Flat-LoRA and LoRA with two SAM's variants, ASAM and GSAM. From the results in Table B2, we observe that optimizing the sharpness over the full parameter space $\mathbf{W}$ generally provides better performance than on the LoRA space $\mathbf{A}$ and $\mathbf{B}$.

Table B2: Comparison with other SAM's variants.

| T5-base | Flat Space | CoLA | MRPC | Extra Memory | Time |
|---|---|---|---|---|---|
| LoRA | - | $82.87_{\pm 0.59}$ | $88.03_{\pm 0.14}$ | - | 1x |
| LoRA+ASAM | A, B | $82.56_{\pm 0.34}$ | $88.09_{\pm 0.27}$ | $\mathcal{O}((m+n) \times r)$ | 2x |
| LoRA+ASAM | W | $83.38_{\pm 0.25}$ | $88.90_{\pm 0.54}$ | $\mathcal{O}(m \times n)$ | 2x |
| LoRA+GSAM | A, B | $82.71_{\pm 0.15}$ | $87.71_{\pm 0.23}$ | $\mathcal{O}((m+n) \times r)$ | 2x |
| LoRA+GSAM | W | $83.77_{\pm 0.45}$ | $89.02_{\pm 0.24}$ | $\mathcal{O}(m \times n)$ | 2x |
| Flat-LoRA | A, B | $83.19_{\pm 0.70}$ | $88.81_{\pm 0.51}$ | $\mathcal{O}(m + r)$ | 1x |
| Flat-LoRA | W | $83.61_{\pm 0.38}$ | $89.59_{\pm 0.37}$ | $\mathcal{O}(m)$ | 1x |

## C    RESULTS ON SUPERGLUE

To further evaluate the effectiveness of our approach, we experiment on more challenging Super-GlUE datasets (Wang et al., 2019a) with T5-base. The training settings remain the same as described in Section 4.1. From the results in Table C3, we observe that Flat-LoRA significantly outperforms LoRA, achieving an average improvement of 1.45% over LoRA.

Table C3: Results (%) on fine-tuning T5-base with a subset of SuperGLUE datasets.

| Datasets | BoolQ | CB | COPA | RTE | WIC | Avg |
|---|---|---|---|---|---|---|
| Full FT | $71.19_{\pm 0.34}$ | $92.86_{\pm 0.13}$ | $66.00_{\pm 1.41}$ | $84.84_{\pm 0.28}$ | $70.38_{\pm 0.36}$ | 77.05 |
| LoRA | $71.61_{\pm 0.41}$ | $92.85_{\pm 0.46}$ | $63.67_{\pm 0.47}$ | $83.03_{\pm 0.26}$ | $67.95_{\pm 0.08}$ | 75.82 |
| Flat-LoRA | $\mathbf{72.62}_{\pm 0.78}$ | $\mathbf{93.75}_{\pm 0.10}$ | $\mathbf{67.00}_{\pm 0.82}$ | $\mathbf{84.48}_{\pm 0.23}$ | $\mathbf{68.50}_{\pm 0.15}$ | $\mathbf{77.27}$ |

## D    MEMORY COST

We report the memory usage for fine-tuning GSM8K datasets using Llama 2-7B model. The experiments are conducted with BF16 mixed-precision training and a micro-batch size of 2, running on an NVIDIA GeForce RTX 4090 GPU. We implement based on our default random seed approach. From the results in Table D4, we observe that Flat-LoRA brings very little additional memory cost compared to LoRA, confirming its effectiveness on maintaining memory efficiency.

## E    ABLATION ON THE VARIANCE MAGNITUDE

To evaluate the impact of perturbation variance, we vary $\sigma^2$ on fine-tuning CIFAR-10/100 with CLIP ViT-B/32. From the results in Table E5, we find that the optimal results are achieved when $\sigma^2$ is 0.10 or 0.15.

Table D4: Comparison on memory usage.

| Method | Memory |
|--------|--------|
| LoRA | 23.49GB |
| Flat-LoRA | 23.61GB |

Table E5: Performance results on CIFAR-10 and CIFAR-100 with different $\sigma^2$ values.

| $\sigma^2$ | 0.01 | 0.05 | 0.10 | 0.15 | 0.20 |
|-----------|------|------|------|------|------|
| CIFAR-10 | 97.92 | 98.02 | 98.05 | 98.09 | 97.74 |
| CIFAR-100 | 88.14 | 88.37 | 88.65 | 88.64 | 88.06 |

## F    RESULTS ON THE CORRUPTION DATASETS

For the corrpution datasets, we fine-tune CLIP ViT-B/32 on CIFAR-100 and test the model on OOD CIFAR-100-C datasets. We report the results across different corruption levels (from 1 to 5). As shown in Table F6, we observe that Flat-LoRA outperforms LoRA more as the corruption level increases. This shows that a flatter local optimum could enhance out-of-domain generalization.

Table F6: Performance comparison of LoRA and Flat-LoRA across different corruption levels. Values in parentheses indicate the improvements of Flat-LoRA over LoRA.

| Corruption Level | 1 | 2 | 3 | 4 | 5 |
|------------------|---|---|---|---|---|
| LoRA | 77.51 | 71.20 | 65.10 | 58.50 | 48.28 |
| Flat-LoRA | 78.89 (**+1.38**) | 73.47 (**+2.27**) | 67.93 (**+2.83**) | 61.54 (**+3.04**) | 51.84 (**+3.56**) |

## G    RESULTS ON SQUAD, XSUM AND CNN/DAILYMAIL

We then conduct experiments on SQuAD (Rajpurkar et al., 2016), XSum (Narayan et al., 2018) and CNN/Dailymail (See et al., 2017) datasets using T5-base model. The training settings remain consistent with those described in Section 4.1 of the paper, except that we adopt 3 training epochs. From the results in Table G7, we observe that Flat-LoRA consistently outperforms LoRA, but the improvement is relatively small. We hypothesize that flatness may not be particularly beneficial for tasks like summarization and question answering but may confer greater advantages for tasks such as mathematics and code-related problems. Moreover, the improvements brought by Flat-LoRA require minimal additional memory and computation cost.

Table G7: Performance comparison between LoRA and Flat-LoRA on different datasets.

| Metric | SQuAD EM/F1 | XSum Rouge1/2/L | CNN/DailyMail Rouge1/2/L |
|--------|-------------|-----------------|--------------------------|
| LoRA | 81.59/89.67 | 34.64/12.36/28.12 | 24.78/12.13/20.51 |
| Flat-LoRA | 81.71/89.84 | 34.88/12.64/28.31 | 24.94/12.26/20.66 |

# H    RESULTS OF MATTHEWS CORRELATION COEFFICIENT ON COLA

In Table 5, we follow the experimental setup of LoRA-GA (Wang et al., 2024), where reports the accuracy metric for CoLA dataset. Here, we evaluate Flat-LoRA using the Matthews Correlation Coefficient (Mcc) metric in Table H8. As shown, again, Flat-LoRA consistently outperforms other methods under the correct Mcc metric.

Table H8: Performance comparison on CoLA dataset.

| Method | Acc | Mcc |
|--------|-----|-----|
| LoRA | $82.87_{\pm 0.22}$ | $59.74_{\pm 1.20}$ |
| AdaLoRA | $82.58_{\pm 0.56}$ | $59.53_{\pm 0.87}$ |
| DyLoRA | $82.98_{\pm 0.34}$ | $59.94_{\pm 1.32}$ |
| Flat-LoRA | $83.61_{\pm 0.38}$ | $61.13_{\pm 1.13}$ |

