# OpenReview forum: "Flat-LoRA: Low-Rank Adaption over a Flat Loss Landscape"
_ICLR.cc/2025/Conference — Submitted to ICLR 2025_

### Official Review · Reviewer_ngBn · 2024-11-02

**Soundness:** 3
**Presentation:** 3
**Contribution:** 3
**Rating:** 6
**Confidence:** 5

**Summary:**

This paper discusses how to improve the generalization of LoRA, with good writing, innovative ideas, and credible experimental performance. However, I think the article overlooks one point: LoRA only fine-tunes a very small number of parameters, which is unlikely to overfit on a specific task and can inherently maintain good generalizability. If I am wrong, please convince me.

**Strengths:**

1. The paper introduces Flat-LoRA, a novel method that improves upon traditional Low-Rank Adaptation (LoRA) techniques by targeting a flat region of the loss landscape. This aims to enhance generalization performance by avoiding sharp minima that can degrade model performance on unseen data.

2. Flat-LoRA addresses the computational and memory cost issues associated with fine-tuning large-scale models. By optimizing only a low-rank matrix and using random weight perturbations, it achieves parameter-efficient fine-tuning without additional memory costs during inference.

3. The paper not only proposes a new technique but also details the underlying mathematical foundations. It discusses the optimization objective, perturbation generation, and integrates a Bayesian expected loss objective to maintain training efficiency.

**Weaknesses:**

1. There is no significant improvement, the limited enhancement on the GLUE benchmark does not prove the lack of generalizability of methods like LoRA.  I need to know if your method is effective on more datasets. More datasets and models should be compared. For example: the SuperGLUE benchmark, SQuAD, XSum, CNN/Dailymail, and some LoRA training on Stable Diffusion.

2. There is a lack of extensive comparisons, such as with methods like DyLoRA[1] and AdaLoRA[2].

3. More relevant articles, such as AWP[3], should be cited, which is an effective method to improve generalization.

[1] Valipour M, Rezagholizadeh M, Kobyzev I, et al. Dylora: Parameter efficient tuning of pre-trained models using dynamic search-free low-rank adaptation[J]. arXiv preprint arXiv:2210.07558, 2022.
[2] Zhang Q, Chen M, Bukharin A, et al. AdaLoRA: Adaptive budget allocation for parameter-efficient fine-tuning[J]. arXiv preprint arXiv:2303.10512, 2023.
[3] Wu D, Xia S T, Wang Y. Adversarial weight perturbation helps robust generalization[J]. Advances in neural information processing systems, 2020, 33: 2958-2969.

**Questions:**

1. How does it perform on the SuperGLUE benchmark, SQuAD, XSum, and CNN/Dailymail?
2. How does it perform on Stable Diffusion?
3. How does it compare between AdaLoRA and DyLoRA?
4. How is the performance on Llama3 with the alpaca dataset?

---

> ### Author Response · Authors · 2024-11-22
> **Response to Reviewer ngBn**
>
> Thank you for your valuable and detailed comments! We are glad that you find our paper good written, with innovative ideas and credible experimental performance. In the following, we provide our point-by-point response and hope our response helps address your concerns. We also look forward to the subsequent discussion which may further help solve the current issues.
>
> **W1: LoRA only fine-tunes a very small number of parameters, which is unlikely to overfit on a specific task and can inherently maintain good generalizability.**
>
> **A1:** Thanks for your insightful question. We agree with you that LoRA is less likely to overfit on a specific task due to its fewer trainable parameters. However, pursuit flatness can still help generation performance even when not in an overfitting regime. To validate this, we perform experiments with LoRA rank=1 and find that Flat-LoRA can still consistently outperform LoRA, although the improvements may not be as large as with a higher rank.
>
> CLIP ViT-B/32 | CIFAR-10 | CIFAR-100
> ---|---|---
> LoRA (rank=1) | 97.64±0.02 | 86.47±0.14
> Flat-LoRA (rank=1) | 97.85±0.05 | 86.70±0.21
>
> We hypothesis that when the lora rank is small, e.g. rank=1, LoRA may attempt to find the lowest loss in the corresponding rank-1 space, which may appear sharp in the full parameter space. This phenomenon is illustrated in the first column of Figure 5 in the paper, where we observe that with a small LoRA rank, the loss landscape in the merged weight space tends to be significantly sharper.
>
> Moreover, the noise injection approach in Flat-LoRA can also be readily incorporated into full fine-tuning to provide significant generalization improvement with minimal additonal memory and computation cost:
> CLIP ViT-B/32 | CIFAR-10 | CIFAR-100
> ---|---|---
> Full-FT | 97.99±0.01 | 89.06±0.11
> Flat-Full-FT | 98.32±0.05 | 90.02±0.08
>
> **W2: The limited enhancement on the GLUE benchmark does not prove the lack of generalizability of methods like LoRA; More datasets and models should be compared. For example: the SuperGLUE benchmark, SQuAD, XSum, CNN/Dailymail, and some LoRA training on Stable Diffusion.**
>
> **A2:** One reason for our limited improvements on GLUE is that we utilize a much higher baseline to evaluate the real effectiveness of our Flat-LoRA, leaving less room for improvement.  In fact, our LoRA baseline is even much better than the optimized LoRA-GA (NeurIPS'24) performance [1], with 88.97 (our LoRA) on MRPC datasets compared to 68.38 (LoRA) and 85.29 (LoRA-GA) reported in [1]. Our improvements on GSM8K and Human-Eval are much significant (i.e. +3.18% and +1.37%) as the tasks are generally harder.
>
> **SuperGLUE**
> We conduct experiments on more challenging SuperGlUE datasets with T5-base. The training settings remain the same as described in Section 4.1 of the paper. The results in the following table shows that Flat-LoRA outperforms LoRA with more significant improvements compared to the results on the GLUE datasets.
> |     T5-base      | BoolQ          | CB             | COPA           | RTE            | WIC            | Avg       |
> | --------- | -------------- | -------------- | -------------- | -------------- | -------------- | --------- |
> | LoRA      | 71.61±0.41     | 92.85±0.46     | 63.67±0.47     | 83.03±0.26     | 67.95±0.08     | 75.82     |
> | Flat-LoRA | **72.62**±0.78 | **93.75**±0.10 | **67.00**±0.82 | **84.48**±0.23 | **68.50**±0.15 | **77.27** |
>
> **SQuAD, XSum, CNN/Dailymail**
> We then conduct experiments on SQuAD, XSum and CNN/Dailymail datasets using T5-base model. The training settings remain consistent with those described in Section 4.1 of the paper, except that we adopt 3 training epochs. From the results below, we observe that Flat-LoRA consisitently outperforms LoRA, but the improvement is relatively small. We hypothesize that flatness may not be particularly beneficial for tasks like summarization and question answering but may confer greater advantages for tasks such as mathematics and code-related problems (see **Q4** below and Section 4.3 in the revision). Moreover, the improvements brought by Flat-LoRA require minimal additional memory and computation cost.
> |   T5-base  | SQuAD | XSum       | CNN/Dailymail |
> | ------ | ----- | ---------- | ------------- |
> | Metric | EM/F1 | Rouge1/2/L | Rouge1/2/L    |
> | LoRA   |  81.59/89.67 |  34.64/12.36/28.12 |     24.78/12.13/20.51  |
> | Flat-LoRA   |   81.71/89.84 |  34.88/12.64/28.31  |    24.94/12.26/20.66    |

---

> ### Author Response · Authors · 2024-11-22
> **Continue with the above response**
>
> **Stable Diffusion**
> We finetune SDXL with the pipeline of Dreambooth with the popular scripts implemented by [HuggingFace](https://github.com/huggingface/diffusers/blob/main/examples/dreambooth/README_sdxl.md).
> As shown in Figure 3 in the revision, Flat-LoRA exhibits better personalization than LoRA while maintaining better generation ability. For instance, in the second column, the image generated by Flat-LoRA includes a distinctive square behind the bird, aligning more closely with the "icon" feature present in the training images (top row). Furthermore, Flat-LoRA more effectively preserves the concept of eyes, whereas in columns 1, 3, and 5, the birds generated by LoRA are missing eyes.
>
> We have included the above experiments into Section 4.4 and Appendix C, G in the revision.
>
>
> [1] Wang, Shaowen, Linxi Yu, and Jian Li. "LoRA-GA: Low-Rank Adaptation with Gradient Approximation." NeurIPS'24.
>
>
> **W3: There is a lack of extensive comparisons, such as with methods like DyLoRA[1] and AdaLoRA[2].**
>
> **A3:** Thanks for your suggestion. We have added the DyLoRA and AdaLoRA baselines in the following table and Table 5 in the revision accordingly.
> Method| CoLA | MRPC |
> ---| ---- | ---- |
> LoRA |82.87±0.22 | 88.03±0.14
> AdaLoRA | 82.58±0.56 | 88.29±0.33
> DyLoRA | 82.98±0.34 | 87.88±0.42
> Flat-LoRA | 83.61±0.38 | 89.59±0.37
>
> **W4: More relevant articles, such as AWP[3], should be cited, which is an effective method to improve generalization.**
>
> **A4:** Thanks for providing the relevant works. We have added into citation accordingly.
>
>
>
> **Q4: Performance on the alpaca dataset**
>
> **A5:** Due to the limited rebuttal time and computation resource, we fine-tune Alpaca dataset with Llama2-13B model to also meet larger LLM experiments requested by Reviewer pg6m. From the results in the following table, we observe that Flat-LoRA consistently outperforms LoRA. We find that the improvements on DROP and HumanEval are more significant (+0.71% and +1.83%), perhaps indicating that flatter minima better support math-related and coding-related tasks. We have added these results into Section 4.3 in the revision.
>
> |Llama2-13B | MMLU  | DROP  | BBH   | HumanEval |
> | ------------------------------------------------------------ | ----- | ----- | ----- | --------- |
> | LoRA                                                         | 51.42 | 37.57 | 34.72 | 13.41     |
> | Flat-LoRA | **51.98** | **38.28** | **34.84** | **15.24**

---

> > ### Comment · Reviewer_ngBn · 2024-11-25
> > **Questions for your performance on CoLA**
> >
> > How about the performance on CoLA for mcc?

---

> > > ### Author Response · Authors · 2024-11-25
> > > **Reply to Reviewer ngBn**
> > >
> > > Thanks for your question regarding mcc performance on CoLA dataset. In our paper, we follow the experimental setup of LoRA-GA [1], which reports results based on the accuracy (acc) metric with T5-base model.
> > >
> > > To address your concern, we have re-evaluated our method using the mcc metric on CoLA dataset, with the results detailed in the following table. As shown, Flat-LoRA consistently outperforms other methods under mcc metric, further confirming the robustness and effectiveness of our approach.
> > >
> > > Method    | CoLA(acc)  | CoLA(mcc) |
> > > ---       |      ----  | ----      |
> > > LoRA      | 82.87±0.22 | 85.84±0.06
> > > AdaLoRA   | 82.58±0.56 | 85.62±0.31
> > > DyLoRA    | 82.98±0.34 | 86.23±0.17
> > > Flat-LoRA | 83.61±0.38 | 86.65±0.27
> > >
> > > [1] Wang, Shaowen, Linxi Yu, and Jian Li. "LoRA-GA: Low-Rank Adaptation with Gradient Approximation." NeurIPS'24.

---

> > > > ### Comment · Reviewer_ngBn · 2024-11-26
> > > > **How about matthews correlation?**
> > > >
> > > > I want to know the matthews correlation on CoLA.

---

> ### Author Response · Authors · 2024-11-26
> **Matthews correlation on CoLA**
>
> Accodringly, we have evaluated our method using the suggested Matthews Correlation Coefficient metric on CoLA dataset. From the results in the following table, we observe, again, Flat-LoRA consistently outperforms other methods under the the matthews correlation metric. We have included this results in the updated version.
>
> Thanks again for your time and effort.
>
>
> Method    | CoLA(acc)  | CoLA(Matthews Cor. Coeff.) |
> ---       |      ----  | ----      |
> LoRA      | 82.87±0.22 | 59.74±1.20
> AdaLoRA   | 82.58±0.56 | 59.53±0.87
> DyLoRA    | 82.98±0.34 | 59.94±1.32
> Flat-LoRA | 83.61±0.38 | 61.13±1.13

---

### Official Review · Reviewer_pg6m · 2024-11-03

**Soundness:** 2
**Presentation:** 3
**Contribution:** 2
**Rating:** 5
**Confidence:** 4

**Summary:**

This paper introduces Flat-LORA, which adds noise to the training process: W + AB + ε, where ε is stored using a seed. The authors attempt to achieve SAM-like effects through this method.

Advantages:
1. The paper is written clearly
2. The core idea seems reasonable

Disadvantages:
1. The paper lacks mathematical rigor in several places, particularly in equations 8 and 9
2. Insufficient experimental validation:
   - Should test on larger models (e.g., LLAMA 13B or 70B)
   - Should evaluate on SuperGLUE
3. Lacks necessary ablation studies, particularly regarding σ^2

Technical Issues:
1. Equations 8 and 9 have fundamental problems:
   - var(X) should be a covariance matrix
   - var(W'_{i,:}X) should be a scalar
   - These dimensions are inconsistent and cannot be equated
2. In Equation 7, n should be sqrt(n), as large n values would result in negligibly small epsilon values added to the weight matrix

Additional Concerns:
1. No memory usage results are reported
2. Table 3 lacks full-tuning baseline results
3. this paper should report results on more diverse datasets
4. More comprehensive ablation studies on σ^2 are needed

**Strengths:**

Advantages:

The paper is written clearly
The core idea seems reasonable

**Weaknesses:**

Disadvantages:

The paper lacks mathematical rigor in several places, particularly in equations 8 and 9
Insufficient experimental validation:
Should test on larger models (e.g., LLAMA 13B or 70B)
Should evaluate on SuperGLUE
Lacks necessary ablation studies, particularly regarding σ^2
Technical Issues:

Equations 8 and 9 have fundamental problems:
var(X) should be a covariance matrix
var(W'_{i,:}X) should be a scalar
These dimensions are inconsistent and cannot be equated
In Equation 7, n should be sqrt(n), as large n values would result in negligibly small epsilon values added to the weight matrix
Additional Concerns:

No memory usage results are reported
Table 3 lacks full-tuning baseline results
this paper should report results on more diverse datasets
More comprehensive ablation studies on σ^2 are needed

**Questions:**

no

---

> ### Author Response · Authors · 2024-11-22
> **Response to Reviewer pg6m**
>
> Thank you for your valuable and detailed comments! We are glad that you find our paper well written with reasonable core idea. In the following, we provide our point-by-point response and hope our response helps address your concerns. We also look forward to the subsequent discussion which may further help solve the current issues.
>
> **W1: The paper lacks mathematical rigor in several places, particularly in equations 8 and 9:
> (1) var(X) should be a covariance matrix;
> (2) var(W'_{i,:}X) should be a scalar;
> (3) These dimensions are inconsistent and cannot be equated**
>
> **A1:** Thank you for pointing out these issues. In equations 8 and 9, we aim to characterize the effects of the varianace of the activation after adding random weight perturbations. We assumed activations have a similar magnitude of variance and used var(X) to represent it. In the revision, we use $\text{var}(\mathbf{x}\_i)$ for clearer notation as a scalar describing the variance of each element. We also rewrite var(W'\_{i,:}X) to $\text{var}(\mathbf{W}'\_{j,:}\mathbf{x})=(1+\sigma^2) \|\mathbf{W}'\_{j, :}\|_2^2\cdot \mathrm{var}(\mathbf{x}\_i)$ to ensure that the output is a scalar. We hope our revision (equations 8-12) could make the statement more rigor.
>
> **W2: Insufficient experimental validation:
> (1) Should test on larger models (e.g., LLAMA 13B or 70B)
> (2) Should evaluate on SuperGLUE**
>
> **A2:**
> Following your suggestions, we fine-tune Llama-2-13B model on Alpaca and evaluate InstructEval metrics. From the results in the following table, we observe that Flat-LoRA consistently outperforms LoRA. We find that the improvements on DROP and HumanEval are more significant (+0.71% and +1.83%), perhaps indicating that flatter minima better support math-related and coding-related tasks.
>
> | Llama2-13B | MMLU      | DROP      | BBH       | HumanEval |
> |  --- |  --- |  --- |  --- |  --- |
> | LoRA       | 51.42     | 37.57     | 34.72     | 13.41     |
> | Flat-LoRA  | **51.98** | **38.28** | **34.84** | **15.24** |
>
> We then evaluate Flat-LoRA on more challenging SuperGlUE datasets with T5-base. The training settings remain the same as described in Section 4.1 of the paper. The results in the following table shows that Flat-LoRA outperforms LoRA with more significant improvements compared to the results on the GLUE datasets.
>
> |   T5-base  | BoolQ          | CB             | COPA           | RTE            | WIC            | Avg       |
> | --- |  --- |  --- |  --- |  --- |  --- |  --- |
> | LoRA      | 71.61±0.41     | 92.85±0.46     | 63.67±0.47     | 83.03±0.26     | 67.95±0.08     | 75.82     |
> | Flat-LoRA | **72.62**±0.78 | **93.75**±0.10 | **67.00**±0.82 | **84.48**±0.23 | **68.50**±0.15 | **77.27** |
>
> We have added these results into Section 4.3 and Appendix C in the revision.
>
> **W3: Lacks necessary ablation studies, particularly regarding σ^2**
>
> **A3:**
> Thank you for your suggestion. We vary $\sigma^2$ on fine-tuning CIFAR-10/100 with CLIP ViT-B/32. We find that the optimal results are achieved when $\sigma^2$ is 0.10 or 0.15. We have included it into Appendix E.
>
> $\sigma^2$| 0.01 | 0.05 | 0.10 | 0.15 | 0.20
> --|--|--|--|---|---
> CIFAR-10 | 97.92 | 98.02 | 98.05 | **98.09** | 97.74
> CIFAR-100 | 88.14 | 88.37 | **88.65** | 88.64 | 88.06
>
>
> **W4: In Equation 7, n should be sqrt(n), as large n values would result in negligibly small epsilon values added to the weight matrix**
>
> **A4:**
> We apologize for the error that the factor in Line 208 should be $1/\sqrt{n}$ rather than $n$, and we have corrected it in the revision.
> We agree that the magnitude should be scaled by a factor of $1/\sqrt{n}$ given the input dimension $n$ to avoid negligibly small $\boldsymbol{\epsilon}$ with large $n$.
> In Equation 7, $n$ represents the order of the variance rather than the standard deviation, and the factor should be squred as $n$. In Equation 8-12 of the revision, we also show that such scaling would increase the variance of the activation by a factor of $\sigma^2$, which is not related to the input dimension $n$ as we expected.
>
> **W5: No memory usage results are reported; Table 3 lacks full-tuning baseline results**
>
> **A5:** We report the memory usage for fine-tuning GSM8K datasets using Llama 2-7B model. The experiments are conducted with BF16 mixed-precision training and a micro-batch size of 2, running on an NVIDIA GeForce RTX 4090 GPU. We observe that Flat-LoRA brings very little additonal memory cost compared to LoRA, comfirming its effectiveness on maintaining memory efficiency. We have included it into Appendix D.
>
> llama2-7b | Memory
> ---|---
> LoRA | 23.49GB
> Flat-LoRA| 23.61GB
>
> We complement the full fine-tuning baselines for Table 3 as follows: when the gap between LoRA and full FT is large, Flat-LoRA can bring more significant improvement over LoRA.
>
> Method | GSM8K | Human-Eval
> --- | -- | ---
> Full FT | 59.36±0.85 | **35.31**±2.13
> LoRA (r=8) | 57.47±0.35 | 24.85±0.52
> Flat-LoRA (r=8) | **60.65**±0.23 | 26.22±0.79

---

> > ### Comment · Reviewer_pg6m · 2024-11-26
> > **I'm still struggling with the rigor of math.**
> >
> > Thank the authors for the response.  I'm still struggling with the rigor of math.
> >
> > I'm a little embarrassed by the way this is written $L(\boldsymbol{W})$. Without input and label ($\boldsymbol{x}, \boldsymbol{y}$), we do not have loss, instead, I would like it written as $L(\operatorname{f}(\boldsymbol{x};\boldsymbol{W}), \boldsymbol{y})$
> >
> >
> > I read again the equations in the paper, and find the paper is still not rigor in mathematic.
> >
> > 1. Eq. 8 seems wrong because if the expectation of $x_i$ is not zero, Eq. 8 is wrong.  Eq. 8 is the foundation of the following equations.
> >
> > I will keep my previous score unchanged.

---

> > > ### Author Response · Authors · 2024-11-26
> > > **Reply to the rigor of math**
> > >
> > > Thank you very much for your careful reading and feedback.
> > >
> > > **Written of $L(\mathbf{W})$**:
> > >
> > > We meant to emphasize the optimization variable $\mathbf{W}$ and thus simply write the loss as $L(\mathbf{W})$. Now we agree with you that it is better to indicator the role of label and input $(\boldsymbol{x},\boldsymbol{y})$, i.e., by using $L(f(\boldsymbol{x};\mathbf{W}), \boldsymbol{y})$. In the revised version, we have explicitly written this notation before abbreviating it to $L(\mathbf{W})$ (Line 156-158).
> > >
> > > **The expectation of $\mathbf{x}_i$ in Eq. 8**:
> > >
> > > Thank you for pointing out the issue when the expectation of $\mathbf{x}\_i$ is not zero. The goal of our analysis in Eq. 8-12 is to justify the use of scaling factor for the standard variation in noise generation, which, as you pointed outed in **W4**, should be $1/\sqrt{n}$ eliminate dependence on the input dimension $n$. The issue does not lie in Eq. 8, which is correct since $\mathbf{W}'\_{j,:}$ is constant. But the deviation of Eq. 11 misses an item related to the expectation of $\mathbf{x}\_i$.
> > > The corrected form can be found in the updated file. The claim that the resulting increased variance is independent of the input dimension $n$ is still true, which aligns with our intended goal.
> > >
> > >
> > > Thank you again for your valuable comments, which have greatly helped improve our manuscript.

---

> > > > ### Author Response · Authors · 2024-11-29
> > > > **We are willing to address your further concerns before the discussion ending**
> > > >
> > > > Dear Reviewer pg6m,
> > > >
> > > > We would like to extend our sincere gratitude for your insightful and valuable comments, which have greatly enriched our manuscript with more experiments on larger model and diverse datasets, improved mathematical rigors, and necessary ablation studies.
> > > >
> > > > The primary contribution of this paper is to propose an efficient and practical approach to pursuing flat minima in the full parameter space for LoRA fine-tuning.
> > > > We try to explain the choice of the scaling factor for noise generation by analyzing the changing behavior of the activations w.r.t. the input dimension $n$. Thanks for your discussion that now we better model and describe the impact of noise injection.
> > > >
> > > > As the discussion period nears its conclusion, we respectfully ask if there are any remaining concerns regarding the mathematical rigor after our revision, and we are willing to engage in further discussions and will do our best to address your concerns ASAP. Thank you!
> > > >
> > > > Best regards,
> > > >
> > > > Paper 8654 Authors

---

> > > > > ### Comment · Reviewer_pg6m · 2024-11-29
> > > > > **Thank you for your reply**
> > > > >
> > > > > Thank the authors for the reply.
> > > > >
> > > > > I read the equation parts, but I still see some mistakes.
> > > > >
> > > > > For example, Eq. 11 misses a square in the second term, and also it also lacks a square in Eq. 12.
> > > > >
> > > > > Eq. 7 is also weird, $\boldsymbol{\epsilon}$ is a vector, 0 in the distribution part is a scalar, and the variance part is a m*n matrix, what distribution does $\boldsymbol{\epsilon}$ satisfy?
> > > > >
> > > > > As an optimization paper, it lacks a mathematical rigourness.

---

> ### Author Response · Authors · 2024-11-29
> **Reply to your further feedback**
>
> Thank you very much for your thorough review and prompt feedback. We address your further concerns below:
>
> **A missing square in Eq. 11&12:**
>
> Indeed, they should be:
>
> Eq. 11:
>
> $$=||\mathbf{W}'\_{j, :}||\_2^2\cdot \mathrm{var}[\mathbf{x}\_i]+n\cdot \frac{\sigma^2}{n} ||\mathbf{W}'\_{j, :}||^2\_2\cdot \left(\mathrm{var}[\mathbf{x}\_i]+\mathbb{E}^2[\mathbf{x}\_i]\right)$$
>
> Eq. 12:
>
> $$
> =(1+\sigma^2) ||\mathbf{W}'\_{j, :}||\_2^2\cdot \mathrm{var}[\mathbf{x}\_i]+\sigma^2||\mathbf{W}'\_{j, :}||^2\_2\cdot\mathbb{E}^2[\mathbf{x}\_i].
> $$
>
> **$\boldsymbol{\epsilon}$ in Eq. 7:**
>
> $\boldsymbol{\epsilon}$ is defined as a matrix with the same dimensions as $\mathbf{W}$, representing the noise added to $\mathbf{W}$. As you correctly point out, the scalar $0$ in Eq. 7 is should be $\mathbf{0}$. To clarify this further, we rewrite the definition of the distribution for each element of $\boldsymbol{\epsilon}$ as follows:
>
> $$\boldsymbol{\epsilon}\_{i,j} \sim \mathcal{N}\left (0, \frac{\sigma^2}{ {n} }   ||\mathbf{W}'\_{i, :}||_2^2 \right).$$
>
> To avoid any misunderstanding that $\boldsymbol{\epsilon}$ represents a vector, we would replace it with notation $\bf N$ as a random matrix in the revised manuscript.
>
> Thank you again for your valuable feedback, which have greatly helped improve our manuscript. We will incorporate these corrections into the revised version.

---

### Official Review · Reviewer_t8En · 2024-11-03

**Soundness:** 3
**Presentation:** 4
**Contribution:** 3
**Rating:** 6
**Confidence:** 5

**Summary:**

Based on the consideration of local optimal values, this paper proposes that the flat optimal value learned by LoRA is not necessarily flat at full rank. Corresponding mathematical explanations are provided. Following this, using the idea of SAM, a method is designed to add full-rank noise perturbations to search for a global optimal solution.

**Strengths:**

1. The writing is well done, with motivations and insights clearly explained in an intuitive manner, accompanied by reasonable mathematical assumptions to introduce the design of full-rank noise perturbations.
2. The design for storing random seeds for memory efficiency and integrating into mixed-precision training is very clever, saving additional overhead.

**Weaknesses:**

1. On line 125, the sentence "the LoRA matrices. better initialization strategy for LoRA parameters." should probably use a comma instead of a period.
2. The work on adding full-rank noise has been done in Noisetune ([https://arxiv.org/abs/2202.12024](https://arxiv.org/abs/2202.12024)) and LORASC ([https://arxiv.org/abs/2407.01491](https://arxiv.org/abs/2407.01491)), especially in LoRASC, where the exact same approach is used, adding a full rank noise to each LoRA optimization process.
3. Overall, the core technical implementation of this paper is to add random perturbations at each step of LoRA training. While the approach is elegant, it’s not particularly novel, and there doesn’t seem to be very convincing experimental results. The proposed Flat-LoRA series offers limited improvement across various LoRA variants and tasks, with most gains being around a few tenths of a percentage in accuracy (except for significant improvements in gsm8k and Human-Eval). It would be helpful to include tasks like apaca and other SFT tasks. I’m particularly interested in the practical significance of this work—flat local optima should theoretically bring stronger generalization, such as supporting OOD (e.g., Alpaca, instruct-eval, etc.) and robustness (e.g., image-R, image-C) evaluations, proving that a flatter local optimum could enhance out-of-domain generalization. This would be valuable since even sharp in-domain optima can perform well, and this might be why performance improvements are modest.

**Questions:**

What do you think is the essential difference or advantage between adding perturbations to weights and adding perturbations to data samples? Is adding perturbations to data samples simpler?

---

> ### Author Response · Authors · 2024-11-22
> **Response to Reviewer t8En**
>
> Thank you for your valuable and detailed comments! We are glad that you find our paper well written with clearly explained insights, and that you consider the idea of storing random seeds for memory efficiency and integrating into mixed-precision training to be very clever. In the following, we provide our point-by-point response and hope our response helps address your concerns. We also look forward to the subsequent discussion which may further help solve the current issues.
>
> **W1: On line 125, the sentence "the LoRA matrices. better initialization strategy for LoRA parameters." should probably use a comma instead of a period.**
>
> **A1:** Thanks for your careful reading. We have fixed the sentence to "initialize the LoRA matrices, which prvoides a better initialization for LoRA parameters".
>
> **W2: The work on adding full-rank noise has been done in Noisetune (https://arxiv.org/abs/2202.12024) and LORASC (https://arxiv.org/abs/2407.01491), especially in LoRASC, where the exact same approach is used, adding a full rank noise to each LoRA optimization process.**
>
> **A2:** Thank you for providing the relevant works; we have included them in our citations. Here, we clarify the differences between our Flat-LoRA and Noisetune and LoRASC:
> 1) **Output**: Flat-LoRA produces a task-specific LoRA for each task, enabling efficient deployment. In contrast, both Noisetune and LoRASC generate full-rank weights for each task. In LoRASC, LoRA parameters are merged into the weights each epoch, resulting in high-rank updates.
> 2) **Objective**: We consider the flatness of weights after merging the LoRA parameters, while LoRASC generates noise based solely on LoRA parameters. Flat-LoRA is more aware of the loss landscape for the final output weights.
> 3) **Method**: We remove the weight noise after calculating the perturbed gradient for each step. However, Noisetune does not require recovering unperturbed weights and adds weight noise only once before training, while LORASC injects once before each training epoch. Moreover, we carefully consider the width factor $n$ into noise generation, and analyze the effects to activations.
> 6) **Memory**: We address memory issues when introducing weight noise and propose strategies to manage it carefully. In comparison, LoRASC needs to store a copy of perturbed weights $\widetilde{W}$, which is not memory efficient for PEFT fine-tuning.

---

> ### Author Response · Authors · 2024-11-22
> **Continue with the above response**
>
> **W3: (1) The proposed Flat-LoRA series offers limited improvement across various LoRA variants and tasks, with most gains being around a few tenths of a percentage in accuracy (except for significant improvements in gsm8k and Human-Eval). It would be helpful to include tasks like apaca and other SFT tasks. (2) I’m particularly interested in the practical significance of this work—flat local optima should theoretically bring stronger generalization, such as supporting OOD (e.g., Alpaca, instruct-eval, etc.) and robustness (e.g., image-R, image-C) evaluations, proving that a flatter local optimum could enhance out-of-domain generalization.**
>
>
> **A3:** Thank you for your detailed suggestions.
> (1) We appreciate your recognition of our significant improvements on GSM8K and Human-Eval. One reason for our limited improvements on GLUE is that we utilize a much higher baseline to evaluate the real effectiveness of our Flat-LoRA, leaving less room for improvement.  In fact, our LoRA baseline is even much better than the optimized LoRA-GA (NeurIPS'24) performance [1], with 88.97 (our LoRA) on MRPC datasets compared to 68.38 (LoRA) and 85.29 (LoRA-GA) reported in [1].
>
>
> To further evaluate the effectiveness of our approach, we conduct experiments on more challenging SuperGlUE datasets with T5-base. The training settings remain the same as described in Section 4.1 of the paper. The results in the following table shows that Flat-LoRA outperforms LoRA with more significant improvements compared to the results on the GLUE datasets.
>
> |     T5-base  | BoolQ          | CB             | COPA           | RTE            | WIC            | Avg       |
> | --------- | -------------- | -------------- | -------------- | -------------- | -------------- | --------- |
> | LoRA      | 71.61±0.41     | 92.85±0.46     | 63.67±0.47     | 83.03±0.26     | 67.95±0.08     | 75.82     |
> | Flat-LoRA | **72.62**±0.78 | **93.75**±0.10 | **67.00**±0.82 | **84.48**±0.23 | **68.50**±0.15 | **77.27** |
>
> (2)
> Following your suggestion, we evaluate OOD and robustness performance on Alpaca and corruption datasets, respectively.
>
> For Alpaca, we fine-tune with Llama-2-13B model and evaluate InstructEval metrics. From the results in the following table, we observe that Flat-LoRA consistently outperforms LoRA. We find that the improvements on DROP and HumanEval are more significant (+0.71% and +1.83%), perhaps indicating that flatter minima better support reasoning and coding tasks.
>
>
> |Llama2-13B | MMLU  | DROP  | BBH   | HumanEval |
> | ------------------------------------------------------------ | ----- | ----- | ----- | --------- |
> | LoRA                                                         | 51.42 | 37.57 | 34.72 | 13.41     |
> | Flat-LoRA | **51.98** | **38.28** | **34.84** | **15.24**
>
> For the corrpution datasets, we fine-tune CLIP ViT-B/32 on CIFAR-100 and test the model on OOD CIFAR-100-C datasets. We report the results across different corruption levels (from 1 to 5). As shown in the following table, we observe that Flat-LoRA outperforms LoRA more as the corruption level increases.  This shows that a flatter local optimum could enhance out-of-domain generalization, as you suggested.
>
> | Corruption level | 1 | 2 | 3 | 4 | 5 |
> | --- | --- | --- | --- | --- | --- |
> |LoRA |  77.51| 71.20| 65.10| 58.50| 48.28
> |Flat-LoRA |  78.89 (**+1.38**)| 73.47 (**+2.27**)| 67.93 (**+2.83**)| 61.54 (**+3.04**)| 51.84 (**+3.56**)
>
> We have added these results into Section 4.3 and Appendix C and F in the revision.
>
> [1] Wang, Shaowen, et al. "LoRA-GA: Low-Rank Adaptation with Gradient Approximation." NeurIPS'24.
>
> **Q1: What do you think is the essential difference or advantage between adding perturbations to weights and adding perturbations to data samples? Is adding perturbations to data samples simpler?**
>
> **A4:** Adding perturbations to weights encourages the network to locate at a flatter minima, leading to better generalization performance. In contrast, adding perturbations to data samples improves the network's robustness to input data perturbation, such as adversarial perturbation in adversarial training. However, adding perturbations to data can lead to the distribution shift of data and ofen results in worse generalization performance on clean data, which is discussed in many adversairal training works, e.g. [1-3].
>
> [1] Raghunathan, Aditi, et al. "Understanding and Mitigating the Tradeoff between Robustness and Accuracy." ICML'20.
>
> [2] Xing, Yue, et al. "On the generalization properties of adversarial training." AISTATIS'21.
>
> [3] Moayeri, Mazda, et al. "Explicit tradeoffs between adversarial and natural distributional robustness." NeurIPS'22.

---

> ### Comment · Area_Chair_honR · 2024-11-30
>
> Dear Reviewer,
>
> Could you kindly respond and indicate whether authors have addressed your concerns?
>
> Thanks, AC

---

> ### Comment · Reviewer_t8En · 2024-11-30
>
> Thank you for the author's response. I understand the differences between the proposed work and Noisetune as well as LORASC mentioned by the author. However, I am not quite clear about the exciting aspects brought by this work. Being different doesn't necessarily constitute a contribution, though I acknowledge that there are some clever ideas in the author's design. Still, I don’t think it represents a significant advancement in understanding for the community.
>
> That said, I find the OOD and robustness experiments to be very positive. At the very least, they demonstrate a certain level of practical significance for this work.
>
> In summary, I will stick to my current rating. I believe this paper is roughly at the level of a weak acceptance.

---

### Official Review · Reviewer_UxAp · 2024-11-04

**Soundness:** 3
**Presentation:** 3
**Contribution:** 3
**Rating:** 6
**Confidence:** 4

**Summary:**

This paper introduces Flat-LoRA, a novel extension to the Low-Rank Adaptation (LoRA) framework, designed to optimize model fine-tuning via discovering solutions within a flatter loss landscape in the full parameter space. Different from traditional LoRA, which may result in sharp solutions that impact generalization negatively, Flat-LoRA incorporates random weight perturbations and a Bayesian expectation loss objective to maintain efficiency. The approach seeks to combine parameter efficiency with enhanced generalization capabilities across both NLP and CV tasks, demonstrating improvements over conventional LoRA in various experimental setups.

**Strengths:**

1. The idea of optimizing to reach a flat landscape in the full parameter space while maintaining the advantages of parameter efficiency is innovative and well-justified.
2. The methodology is generally clearly articulated, and lots of experiments show considerable improvements over existing LoRA-based methods, which validates the efficacy of Flat-LoRA proposed.
3. The explanation of the Bayesian expectation loss objective function and the perturbation generation strategy is very thorough and definitely contributes to the transparency of the proposed method.
4. This work is equipped with practical implications, as fine-tuning LLMs efficiently is increasingly important in current ML field.

**Weaknesses:**

1. Althougth the paper emphasizes the computational overhead and the minimal memory, the perturbation generation and its integration into the mixed-precision training could be simplified or clarified furthermore.
2. The method concentrates mostly on linear layers in the transformer-based models. Despite the fact that the authors acknowledge this as a limitation, extending such approach to other parameter categories would make the method more versatile.
3.: Although the comparisons with approaches such as SAM are very insightful, deeper analysis with more recent variations of sharpness-aware algorithms could strengthen the study and contribution.

**Questions:**

Could the perturbation generation strategy be optimized or adapted to incorporate other noise categories (e.g., adversarial perturbations)?

---

> ### Author Response · Authors · 2024-11-22
> **Response to Reviewer UxAp**
>
> Thank you for your valuable and detailed comments! We are glad that you find our idea of optimizing to reach a flat landscape in the full parameter space to be innovative and well-justfied with practical implications. In the following, we provide our point-by-point response and hope our response could  address your concerns. We also look forward to the subsequent discussion which may further help solve the current issues.
>
> **W1: Althougth the paper emphasizes the computational overhead and the minimal memory, the perturbation generation and its integration into the mixed-precision training could be simplified or clarified furthermore.**
>
> **A1:** Thank you for your suggestion. As you mentioned, the main contribution of this paper is the efficient and effective generation of perturbations to optimize the flatness of loss landscape in the full parameter space. For mixed precision training, which is widely adopted during LLM training, we offer an easy approach to seamlessly integrate the perturbation injection process into the precision casting, introducing no addtional memory cost. However, our main approach is to efficiently store the perturbation based on filter norms and random seed, which is more general and does not require mixed-precision training.
>
> We have made the statement more clear in the revision (Page 5).
>
>
> **W2: The method concentrates mostly on linear layers in the transformer-based models. Despite the fact that the authors acknowledge this as a limitation, extending such approach to other parameter categories would make the method more versatile.**
>
> **A2:** Following your suggestion, we extend the injection of random weight perturbation to all layers, referred to as "Flat-LoRA (all)". Specifically, we additionally add perturbations to layernorm layers, biases, and class embeddings, etc. We generate noise based on the absolute weight $|\mathbf{W}|$. From the results in the following table, we observe that Flat-LoRA (all) indeed improves performance, though the improvement is not as large as Flat-LoRA (Linear) over LoRA. We include this extension in Appendix A.
>
> | CLIP ViT-B/32 | CIFAR-10 | CIFAR-100 |
> ---| -------- | --------- |
> LoRA | 97.90±0.02 | 87.74±0.13
> Flat-LoRA (linear) | 98.09±0.04  | 88.64±0.23 |
> Flat-LoRA (all) |  **98.13**±0.03   |  **88.76**±0.19  |
>
>
> **W3.: Although the comparisons with approaches such as SAM are very insightful, deeper analysis with more recent variations of sharpness-aware algorithms could strengthen the study and contribution.**
>
> **A3:** Following your suggestion, we also compare Flat-LoRA and LoRA with two SAM's variants, ASAM and GSAM. From the results in the following table, we observe that optimizing the sharpness over the full parameter space $\mathbf{W}$ generally provides better performance than on the LoRA space $\mathbf{A}$ and $\mathbf{B}$. We have added these experiments into Appendix B.
>
> T5-base | Flat Space | CoLA | MRPC | Extra Memory | Time
> ---|---|---|---|---|---|
> LoRA| - | 82.87±0.59 | 88.03±0.14 | - | 1x
> LoRA+ASAM | $\mathbf{A},\mathbf{B}$ | 82.56±0.34 | 88.09±0.27 |$\mathcal{O} ((m+n)\times r)$ | 2x
> LoRA+ASAM | $\mathbf{W}$ | 83.38±0.25 | 88.90±0.54 | $\mathcal{O} (m\times n)$ | 2x
> LoRA+GSAM | $\mathbf{A},\mathbf{B}$ | 82.71±0.15 | 87.71±0.23 |$\mathcal{O} ((m+n)\times r)$ |2x
> LoRA+GSAM | $\mathbf{W}$ | **83.77**±0.45 | 89.02±0.24 |$\mathcal{O} (m\times n)$ | 2x
> Flat-LoRA |  $\mathbf{A},\mathbf{B}$ | 83.19±0.70 | 88.81±0.51 | $\mathcal{O}(m+r)$ | 1x
> Flat-LoRA | $\mathbf{W}$ | 83.61±0.38 | **89.59**±0.37 | $\mathcal{O}(m)$ | 1x
>
> **Q1: Could the perturbation generation strategy be optimized or adapted to incorporate other noise categories (e.g., adversarial perturbations)?**
>
> **A4:** Yes. It is possible to incorporate adversarial perturbations with random weight perturbations, i.e., perturbing the weights with random perturbations before applying adversairal ones, to enhance generalization performance, as shown in R-SAM [1].
> It's also possible to alternatively use random and adversarial perturbations to improve the efficiency of sharpness-aware training within a certain computation budget.
> However, we note that incoporating adversarial weight perturbation doubles the time training cost, requiring two gradient step per iteration, and introduces a hard copy of the perturbation weights in memory, which may be undesirable for PEFT training setting.
>
> [1] Liu, Yong, et al. "Random sharpness-aware minimization." Advances in Neural Information Processing Systems (2022).

---

> > ### Comment · Reviewer_UxAp · 2024-11-28
> > **Thanks for the reponse**
> >
> > Thanks for the authors' response which solves most of my concerns, hence I maintain my score.

---

### Author Response · Authors · 2024-11-25
**Looking forward to the feedback**

Dear AC and PC Members,

First and foremost, we would like to express our sincere gratitude for your time and efforts in reviewing our manuscript. As the time is coming to the discussion ending, we are eager to engage in further discussions and do our best to address the reviewers' concerns, which might help you with better decisions. Below, we summarize the reviewers' comments and our corresponding responses:

1. **Limited improvement on GLUE:** we use a stronger baseline compared to previous works (e.g., LoRA-GA) for more rigor evaluation. Additionally, experiments on more challenging SuperGLUE demonstrates greater improvements of Flat-LoRA.
2. **Limited experiments scope:** we have substantially extended our evaluation scopes with larger models (e.g., LLama-2-13B) and various datasets (e.g., Alpaca, SuperGLUE, etc.). We also discuss the effectiveness of flatness across different tasks.
3. **Lack of mathematical rigor:** we have revised the corresponding part to clarify and address potential misunderstandings. We also correct and justify the scaling factor $1/\sqrt{n}$ for noise generation.
4. **Unlikely overfitting for LoRA:** we visualize how very low-rank may appear sharp in the full parameter space and show that flatness can help even in underfitting regime. We also show that the noise injection approach in Flat-LoRA can be readily incorporated into full fine-tuning.
5. **Lack of comparisons, ablations, citations:** we have included these experiments and references accordingly as suggested by the reviewers.


The necessary updates have been integrated into the revised PDF file, with changes highlighted in blue. We sincerely hope the reviewers could provide further feedback, and we look forward to continuing discussions with you.

Best regards,

The Anonymous Author(s) of Paper8654

---

### Comment · Area_Chair_honR · 2024-11-28

Dear Reviewers,

If you have not responded to author's rebuttal, please kindly do so as soon as possible. The deadline is Dec 2, but the authors can potentially further clarify questions if you respond earlier. Thanks!

Best, AC

---

### Meta-Review · Area_Chair_honR · 2024-12-24

**Metareview:**

Summary: Flat-LoRA adds random weight perturbations into LoRA optimization to locate flat minima in the loss landscape. This improves performance and robustness in vision and language models.

Strengths: novel idea considering full parameter space in LoRA; practical method; clear presentation.

Weaknesses: marginal performance gains; missing comparisons with baselines; limited datasets for evaluation;

Reasons for the decision: while Flat-LoRA introduces a novel idea of optimizing flat minima, its marginal performance improvements, limited baselines, and datasets in evaluation do not justify acceptance. In general the paper does not receive enough support from the reviewers.

**Additional Comments On Reviewer Discussion:**

The authors provided additional experiments on larger models (e.g., LLaMA-13B), added baselines (e.g., AdaLoRA), and addressed some mathematical inconsistencies. However, theoretical and experimental limitations, including limited dataset diversity and marginal improvements, remain unresolved.

---

### Decision · Program_Chairs · 2025-01-22

Reject